# Guaranteed Neural PDE Boundary Control with Neural Barrier Function

## Abstract

The physical world dynamics are generally governed by underlying partial derivative equations (PDEs) with unknown analytical forms in science and engineering problems. Neural network based data-driven approaches have been heavily studied in simulating and solving PDE problems in recent years, but it is still challenging to move forward from understanding to controlling the unknown PDE dynamics. PDE boundary control instantiates a simplified but important problem by only focusing on PDE boundary conditions as the control input and output. However, current model-free PDE controllers cannot ensure the boundary output satisfies some given user-specified safety constraint. To this end, we propose a safety filtering framework to guarantee the boundary output stays within the safe set for current model-free controllers. Specifically, we first introduce a general neural boundary control barrier function (BCBF) to ensure the feasibility of the trajectory-wise constraint satisfaction of boundary output. Based on a neural operator modeling the transfer function from boundary control input to output trajectories, we show that the change in the BCBF depends linearly on the change in input boundary, so quadratic programming-based safety filtering can be done for pre-trained model-free controllers. Extensive experiments under challenging hyperbolic, parabolic and Navier-Stokes PDE dynamics environments validate the effectiveness of the proposed method in achieving better general performance and boundary constraint satisfaction compared to the model-free controller baselines.

## 1 Introduction

Partial derivative equations (PDEs) characterize the most fundamental laws of the continuous dynamical systems in the physical world Evans (1998); Perko (1996). Non-analytical PDE dynamics are often involved in complicated science and engineering problems of computational fluid dynamics Kochkov et al. (2021), computational mechanics Samaniego et al. (2020), robotics Heiden et al. (2021), etc. Recently, neural networks have largely boosted the study of numerical PDE solvers using data-driven methods, simulating and characterizing the dynamics Raissi et al. (2019); Brunton & Kutz (2024); Kovachki et al. (2023). However, the PDE control problem still remains challenging without any prior about underlying PDE equations, serving as a huge gap from understanding science to solving engineering problems Yu & Wang (2024).

Recent pioneer works Bhan et al. (2024); Zhang et al. (2024a) provide various formulations of PDE control problems and multiple benchmark settings, either in-domain control Zhang et al. (2024b) or boundary control Bhan et al. (2023). Since it is easier to control the PDE boundary in the real world, following Bhan et al. (2024), we focus on the PDE boundary control setting where the control signal essentially serves as the boundary condition and the unknown PDE dynamics itself remains unchanged. Model-based PDE boundary control has been studied for years, and backstepping-based methods have been applied to different PDE dynamics Krstic & Smyshlyaev (2008b). Nevertheless, the model-based methods cannot work well under the unknown PDE dynamics, suffering from significant model mismatch. Model-free reinforcement learning (RL) controllers Schulman et al. (2017); Haarnoja et al. (2018) have shown impressive results in the benchmark Bhan et al. (2024) compared to the model-based control methods Pyta et al. (2015).

Besides, constraint satisfaction is of great importance for the PDE boundary control problems, but current safe PDE control methods are typically backstepping-based and require knowledge about the

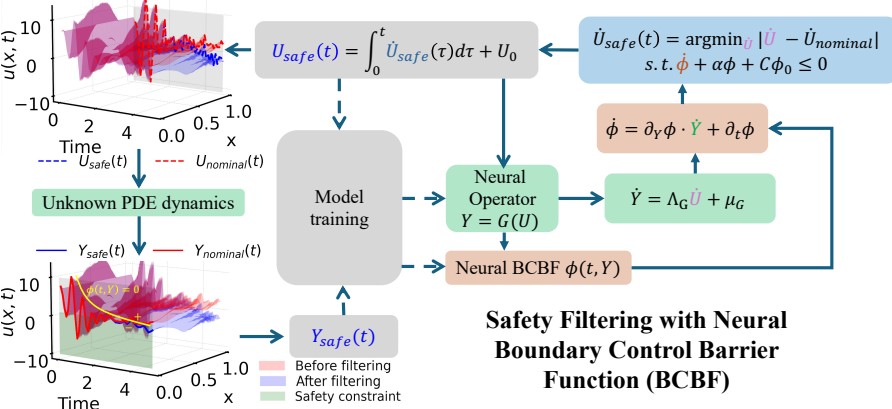

Figure 1: Overview of our safety filtering method for PDE boundary control with neural BCBF. Solid line arrows denote the safety filtering, while dashed ones denote the model training.

PDE dynamics (Krstic & Bement, 2006; Li & Krstic, 2020; Koga & Krstic, 2023; Wang & Krstic, 2023). The constraint considered in this paper is called *boundary feasibility*, which characterizes whether the boundary output falls into and stays within the safe set at the end of the finite-time trajectory, and can be understood as the constraint of finite-time convergence. Under ordinary differential equations (ODEs) setting, neural network parameterized control Lyapunov/barrier functions (CLF/CBFs) have been adopted to ensure the convergence and safety of learning-based controllers Boffi et al. (2021); Dawson et al. (2023); Chang et al. (2019); Mazouz et al. (2022), based on the Markov property of the dynamics at each step , i.e., the change of state only depends on the current state and control input. However, the Markov assumption does not generally hold for PDE boundary control due to infinite-dimensional unobserved states along the spatial axis. Hence, it is challenging to adopt ODE CBFs and find the boundary control input at each step for trajectory-wise convergence over boundary constraint satisfaction in the PDE setting.

To this end, we introduce a new framework to achieve *boundary feasibility* within a given safe set for the PDE boundary control problem, as shown in Figure 1. More specifically, we propose neural boundary control barrier functions (BCBFs) over the boundary output to enable the incorporation of the time variable with a finite-time convergence guarantee. Then, we adopt a neural operator to directly learn the mapping from boundary input to output as a transfer function. Combining well-trained neural BCBF and neural operator, we show a linear dependence between boundary feasibility condition and the derivative of boundary control input, making the safety filtering possible by projecting the actions from the nominal RL controller to the safe boundary control input set using quadratic programming (QP). We conduct experiments on multiple PDE benchmarks and show our superiority over RL controllers in terms of general performance and constraint satisfaction. To the best of our knowledge, we are the first to study the safe boundary control with unknown PDE dynamics. We summarize our contributions below.

- We propose a novel PDE safe control framework with a neural boundary control barrier function to guarantee the boundary feasibility of boundary output within a given safe set.

- We model the control input and output mapping through a neural operator as a transfer function and prove that it can be used for safety filtering by solving quadratic programming.

- We show that the performance after safety filtering performs better compared to the original RL controllers in reward and boundary feasibility rate and time steps on multiple PDE environments.

## 2 PROBLEM FORMULATION

Following the PDE boundary control setting (Bhan et al., 2024), we consider the state $u(x, t) : \mathcal{X} \times \mathcal{T} \to \mathcal{S} \subset \mathbb{R}$ from the continuous function space $C(\mathcal{X} \times \mathcal{T}; \mathbb{R})$ governed by underlying closed-loop partial differential equation (PDE) dynamics defined on normalized $n$-dimensional spatial domain

$\mathcal{X} = [\mathbf{0}, \mathbf{1}] := [0, 1]^n \subset \mathbb{R}^n$ and temporal domain $\mathcal{T} = [0, T] \subset \mathbb{R}^+$ as follows,

$$\frac{\partial u}{\partial t} = \mathcal{D}(u, \frac{\partial u}{\partial x}, \frac{\partial^2 u}{\partial x^2}, \dots, U(t)), x \in \mathcal{X}, t \in \mathcal{T}, u \in \mathcal{S}, \tag{1}$$

where $\mathcal{D}$ is the PDE system dynamics and $U(t)$ is the control signal as the boundary condition. Without loss of generality, we focus on the Dirichlet boundary control input as $U(t) := u(\mathbf{1}, t)$ with constant initial condition $u(x, 0) \equiv U(0) \in \mathcal{S}$. Instead of optimizing boundary input $U(t)$ to track or stabilize full-state observation trajectory $u(x, t)$ (Bhan et al., 2024), we aim to find $U(t)$ that guarantees the *boundary feasibility* of boundary output $Y(t) := u(\mathbf{0}, t)$ within the given user-specified safe set $\mathcal{S}_0 \subset \mathcal{S}$ over $\mathcal{T}$, i.e., $\exists t_0 \in \mathcal{T}, \forall t \geq t_0, Y(t) \in \mathcal{S}_0$. More formally, we give the definition of *boundary feasibility* in PDE dynamics.

**Definition 2.1** (Boundary Feasibility for Trajectory-wise Finite-time Constraint Satisfaction). With state $u(x, t)$ subjected to closed-loop PDE dynamics in Equation (1) with the boundary control input $U(t)$, the boundary control output $Y(t)$ is defined to be feasible over $\mathcal{T}$ within the given user-specified safe set $\mathcal{S}_0 \in \mathcal{S}$ if the following holds,

$$\exists t_0 \in \mathcal{T}, \forall t_0 \leq t \leq T, Y(t) := u(\mathbf{0}, t) \in \mathcal{S}_0, \text{ where } u(\mathbf{1}, t) = U(t), u(x, 0) \equiv U(0). \tag{2}$$

Besides, we adopt the supervised learning scheme with a collected dataset of boundary input and output trajectory pairs $\{[U_k(t), Y_k(t)], k = 1, 2, \dots, K\}$ with sampled discretization from the unknown PDE dynamics. Therefore, we formulate the problem for this paper as follows.

**Problem 2.1.** Given $K$ collected boundary input and output trajectory pairs $\{[U_{k,m}, Y_{k,m}], k = 1, 2, \dots, K, m = 1, 2, \dots, M\}$ with $M$-point temporal discretization, under consistent initial condition $u_k(x, 0) \equiv U_k(0)$ from unknown but time-invariant PDE dynamics in Equation (1), we aim to find boundary control input $U(x)$ that guarantees boundary feasibility of boundary output $Y(t)$ with user-specified safe set $\mathcal{S}_0$ in Definition 2.1.

# 3 METHODOLOGY

## 3.1 NEURAL BARRIER FUNCTION FOR PDE BOUNDARY CONTROL

Control barrier functions (CBFs) are shown to be successful for safe control (Liu & Tomizuka, 2014; Ames et al., 2014) and neural networks have been heavily investigated to effectively parameterize CBFs (Robey et al., 2020; Liu et al., 2022; Zhang et al., 2023) for ODE dynamics. Since the Markov assumption does not hold for PDE boundary control problem, it is challenging to leverage conventional CBF to directly find control input $U$ at time $t$ for the constraint satisfaction of the marginalized output boundary $Y(t) := u(\mathbf{0}, t)$ from the underlying PDE dynamics with spatially-continuous unobserved state $u(x, t)$. To mitigate this issue, inspired by Garg & Panagou (2021b), we propose a more general neural boundary control barrier function (neural BCBF), explicitly incorporating time $t$ into neural network parameterized function $\phi(t, Y) : \mathcal{T} \times \mathcal{S} \to \mathbb{R}$ for the time-dependent zero-sublevel set $\mathcal{S}_{\phi,t} := \{Y(t) \mid \phi(t, Y(t)) \leq 0\}$. Note that the conventional CBF $\phi(Y)$ can be viewed as a specially case of BCBF $\phi(t, Y)$ where $t$ remains constant, so we also regard $\phi(Y)$ as BCBF. Another challenge is that the boundary feasibility in Equation (2) for PDE boundary control is defined on finite time domain $\mathcal{T} = [0, T]$, which requires higher convergence rate to the safe set than the original asymptotic CBF Ames et al. (2014) like fixed-time stability in Polyakov (2011); Garg & Panagou (2021a). We show the following theorem for the feasibility of boundary control output $Y(t)$ within user-specified safe set $\mathcal{S}_0$ under boundary control signal $U(t)$.

**Theorem 3.1** (Boundary Feasibility with Boundary Control Barrier Function). For the state $u(x, t)$ from the closed-loop PDE dynamics with boundary control input $U(t) = u(\mathbf{1}, t), u(x, 0) \equiv U_0$, the boundary feasibility of boundary output $Y(t) = u(\mathbf{0}, t)$ over $\mathcal{T} = [0, T]$ within user-specified safe set $\mathcal{S}_0$ is guaranteed with neural BCBF $\phi(t, Y)$ if the following holds $\forall t \in \mathcal{T}$

$$\left(\mathcal{S}_{\phi,t} := \{Y \mid \phi(t, Y) \leq 0\} \subseteq \mathcal{S}_0\right) \bigwedge \left(\partial_Y \phi \cdot \frac{dY}{dt} + \partial_t \phi + \alpha\phi(t, Y) + C_{\alpha,T}\phi(0, U_0) \leq 0\right), \tag{3}$$

where $C_{\alpha,T} := \frac{\alpha}{e^{\alpha T} - 1} > 0$ is a constant for finite-time convergence. Similarly, the boundary feasibility with neural BCBF $\phi(Y)$ holds if Equation (3) holds under $\partial_Y \phi = \nabla_Y \phi, \partial_t \phi = 0$.

The proof can be found in the Appendix A.2. With the $M$-point temporal discretization of collected boundary input and output trajectory $\{[U_{k,m}, Y_{k,m}], k = 1, 2, \ldots, K, m = 1, 2, \ldots, M\}, \mathcal{S}_{\phi,t} \subseteq \mathcal{S}_0$ in Equation (3) induces the loss function below following Dawson et al. (2022)

$$\mathcal{L}_\mathcal{S} = \sum_{k=1}^K \sum_{Y_{k,m} \in \mathcal{S}_0} [\phi(t_m, Y_{k,m})]_+ + \sum_{k=1}^K \sum_{Y_{k,m} \notin \mathcal{S}_0} [-\phi(t_m, Y_{k,m})]_+, \text{ with } [\cdot]_+ := \max\{0, \cdot\}. \quad (4)$$

However, it is challenging to find $dY(t)/dt$ involved in Equation (3) over the discrete time samples since the boundary output $Y(t) = u(\mathbf{0}, t)$ is governed by the unknown closed-loop PDE dynamics with the boundary condition $U(t) = u(\mathbf{1}, t)$. Besides, it is also non-trivial to find the boundary feasibility condition over boundary control input $U(t)$ for safety filtering due to non-Markov property. Therefore, we adopt the neural operator to learn the boundary input-output mapping as a neural transfer function.

## 3.2 Learning Neural Operator for Input-output Boundary Mapping

Different from current applications of neural operators in learning PDE solutions by temporal mapping Li et al. (2020a;b; 2022), we propose to adopt neural operator $\mathcal{G}_\theta : \{U : \mathcal{T} \to \mathcal{S}\} \mapsto \{Y : \mathcal{T} \to \mathcal{S}\}$ to model the spatial boundary mapping from input to output of the unknown closed-loop PDE dynamics in Equation (1), i.e., $Y(t) = u(\mathbf{1}, t) = \mathcal{G}_\theta(U)(t) = \mathcal{G}_\theta(u(\mathbf{0}, t))(t)$. Following Kovachki et al. (2023) under the setting of same Lebesgue-measurable domain $\mathcal{T}$ for hidden layers, the neural operator is defined as $\mathcal{G}_\theta = \mathcal{Q} \circ \mathcal{I}_{L-1} \circ \cdots \circ \mathcal{I}_0 \circ \mathcal{P}$, including pointwise lifting mapping $\mathcal{P} : \{U : \mathcal{T} \to \mathcal{S}\} \mapsto \{v_0 : \mathcal{T} \to \mathbb{R}^{d_{v_0}}\}$, iterative kernel integration layers $\mathcal{I}_l : \{v_l : \mathcal{T} \to \mathbb{R}^{d_{v_l}}\} \mapsto \{v_{l+1} : \mathcal{T} \to \mathbb{R}^{d_{v_{l+1}}}\}, l = 0, \ldots, L-1$, and the pointwise projection mapping $\mathcal{Q} : \{v_L : \mathcal{T} \to \mathbb{R}^{d_{v_L}}\} \mapsto \{Y : \mathcal{T} \to \mathcal{S}\}$. Specifically, the $l$-th kernel integration layer follows the following form with commonly-used integral kernel operator Li et al. (2020a;b; 2022),

$$v_{l+1}(t) = \mathcal{I}_l(v_l)(t) = \sigma_{l+1}\left(W_l v_l(t) + \int_\mathcal{T} \kappa^{(l)}(t, s) v_l(s) ds + b_l(t)\right), l = 0, 1, \ldots, L-1, \quad (5)$$

where $\sigma_{l+1} : \mathbb{R}^{d_{v_{l+1}}} \to \mathbb{R}^{d_{v_{l+1}}}$ is the activation function, $W_l \in \mathbb{R}^{d_{v_{l+1}} \times d_{v_l}}$ is the local linear operator, $\kappa^{(l)} \in C(\mathcal{T} \times \mathcal{T}; \mathbb{R}^{d_{v_{l+1}} \times d_{v_l}})$ is the kernel function for integration, and $b_l \in C(\mathcal{T}; \mathbb{R}^{d_{v_{l+1}}})$ is the bias function. Besides, since lifting and projection operators $\mathcal{P}, \mathcal{Q}$ are pointwise local maps as special Nemitskiy operators (Dudley et al., 2011; Kovachki et al., 2023), i.e. there exist equivalent functions $P : \mathcal{S} \to \mathbb{R}^{d_{v_0}}, Q : \mathbb{R}^{d_{v_L}} \to \mathcal{S}$ such that $\mathcal{P}(U)(t) = P(U(t)), \mathcal{Q}(v_L)(t) = Q(v_L(t)), \forall t \in \mathcal{T}$. Therefore, combining Equation (5), we explicitly show the boundary mapping from control input $U(t)$ to output $Y(t)$ below, making them possible to be directly connected as $Y(t) = \mathcal{G}_\theta(U)(t)$,

$$Y(t) = \mathcal{G}_\theta(U)(t) = Q(v_L(t)), v_{l+1}(t) = \mathcal{I}_l(v_l)(t) \text{ in Equation (5)}, v_0(t) = P(U(t)), \quad (6)$$

where $P, Q, W_l, \kappa^{(l)}, b_l, l = 0, 1, \ldots, L-1$ parameterized with neural networks $\theta$ and compose the neural operator $Y(t) = G_\theta(U)(t)$. Given boundary input and output $M$-step temporally discretized $K$ trajectory pairs $\{[U_{k,m}, Y_{k,m}], k = 1, 2, \ldots, K, m = 1, 2, \ldots, M\}$, $G_\theta$ and neural BCBF $\phi$ can be optimized together based on empirical-risk minimization using the following loss function,

$$\min_{\theta, \phi} \lambda_\mathcal{G} \mathcal{L}_\mathcal{G} + \lambda_\mathcal{S} \mathcal{L}_\mathcal{S} + \lambda_{BF} \mathcal{L}_{BF}, \text{ where } \mathcal{L}_\mathcal{G} = \sum_{k=1}^K \sum_{m=1}^M \|Y_{k,m} - \mathcal{G}_\theta(U_k)(t_m)\|^2, \mathcal{L}_\mathcal{S} \text{ in eq. (4)},$$

$$\mathcal{L}_{BF} = \sum_{k=1}^K \sum_{m=1}^M [\partial_{Y_{k,m}} \phi \cdot \frac{d\mathcal{G}_\theta(U_k)(t)}{dt}|_{t=t_m} + \partial_{t_m} \phi + \alpha \phi(t_m, Y_{k,m}) + C_{\alpha,T} \phi(0, U_{k,0}), \quad (7)$$

and $[\cdot]_+ := \max\{0, \cdot\}, , \lambda_\mathcal{G}, \lambda_\mathcal{S}, \lambda_{BF}$ are weight hyperparameters for $\mathcal{L}_\mathcal{G}, \mathcal{L}_\mathcal{S}, \mathcal{L}_{BF}$, respectively. The loss for neural operator learning $\mathcal{L}_\mathcal{G}$ is based on Equation (6), and the boundary feasibility (BF) loss of $\mathcal{L}_{BF}$ is based on Equation (3) with the replacement of $dY(t)/dt$ with $d\mathcal{G}_\theta(U)(t)/dt$, which will be detailed in the next section.

## 3.3 Safety Filtering with Quadratic Programming

Once the boundary input-output mapping is modeled by neural operator $\mathcal{G}_\theta$, the boundary output $Y(t)$ is directly related to boundary input $U(t)$ from trajectory to trajectory, bypassing the non-Markov property and the unknown closed-loop dynamics in Equation (1). We first find the derivative

of boundary output $Y(t)$ w.r.t $t$ based on neural operator $Y(t) = \mathcal{G}_\theta(U)(t)$. Applying chain rule to Equation (6), the following derivatives hold,

$$\frac{dY(t)}{dt} = \nabla Q^\top \frac{dv_L(t)}{dt}, \frac{dv_{l+1}(t)}{dt} = \mathcal{J}_l(\frac{dv_l}{dt})(t), \text{ for } l = L-1, \ldots, 0, \frac{v_0(t)}{dt} = \nabla P^\top \frac{dU(t)}{dt}, \quad (8)$$

where the derivative of kernel integration layer $\mathcal{J}_l : \{\frac{v_l}{dt} : \mathcal{T} \to \mathbb{R}^{d_{v_l}}\} \mapsto \{\frac{v_{l+1}}{dt} : \mathcal{T} \to \mathbb{R}^{d_{v_{l+1}}}\}, l = 0, 1, \ldots, L-1$ can be found through the derivative of Equation (5) in a recursive form below,

$$\frac{dv_{l+1}(t)}{dt} = \mathcal{J}_l(\frac{dv_l}{dt})(t) = \text{Diag}(\sigma'_{l+1}) \left( W_l \frac{dv_l(t)}{dt} + \int_\mathcal{T} \frac{\partial \kappa^{(l)}(t,s)}{\partial t} v_l(s) ds + \frac{db_l(t)}{dt} \right). \quad (9)$$

By combining Equation (8) and Equation (9), we have the following theorem to show how the boundary control input $U(t)$ can be chosen to guarantee the boundary feasibility of boundary output $Y(t)$ modeled by neural operator $\mathcal{G}_\theta$.

**Theorem 3.2** (Boundary Feasibility with Neural Operator). Assuming the neural operator $\mathcal{G}_\theta$ as an exact map from boundary input $U(t)$ to output $Y(t)$ for an unknown closed-loop PDE dynamics without model mismatch, the boundary control input $U(t)$ is guaranteed to induce boundary feasibility of output $Y(t)$ over $\mathcal{T} = [0, T]$ within the sublevel set of neural BCBF $\phi$ if $U(t)$ satisfies

$$\partial_Y \phi(t, \mathcal{G}_\theta(U)) \frac{d\mathcal{G}_\theta(U)(t)}{dt} + \partial_t \phi(t, \mathcal{G}_\theta(U)) + \alpha \phi(t, \mathcal{G}_\theta(U)) + C_{\alpha,T} \phi(0, U(0)) \leq 0, \forall t \in \mathcal{T} \quad (10)$$

where $C_{\alpha,T} = \frac{\alpha}{e^{\alpha T}-1}$, and $\frac{d\mathcal{G}_\theta(U)(t)}{dt}$ can be found below with $\prod_1^0(\cdot) := 1$,

$$\frac{d\mathcal{G}_\theta(U)(t)}{dt} = \nabla Q^\top \prod_{l=0}^{L-1} \left( \text{Diag}(\sigma'_{L-l}) W_{L-1-l} \right) \nabla P^\top \frac{dU(t)}{dt} + \nabla Q^\top \text{Diag}(\sigma'_L) \sum_{i=0}^{L-1} \left( \left[ \prod_{j=1}^{i} W_{L-j} \right] \right.$$

$$\left. \text{Diag}(\sigma'_{L-j}) \right] \left( \int_\mathcal{T} \frac{\partial \kappa^{(L-1-i)}(t,s)}{\partial t} v_{L-1-i}(s) ds + \frac{db_{L-1-i}(t)}{dt} \right) \right) = \Lambda_\theta(t)\dot{U}(t) + \mu_\theta(t). \quad (11)$$

*Remark.* We remark that if the sublevel set of neural BCBF $\phi$ is a subset of user-specified safe set $\mathcal{S}_0$, and there is no model mismatch between neural operator $Y(t) = \mathcal{G}_\theta(U)(t)$ and unknown closed-loop PDE dynamics, Theorem 3.2 is equivalent to Theorem 3.1. Then the boundary control input $U(t)$ satisfying Equation (10) is guaranteed to induce the boundary feasibility of boundary output $Y(t)$ within user-specified safe set $\mathcal{S}_0$. Similarly, Theorem 3.2 with neural BCBF $\phi(Y)$ holds if Equation (10) holds by letting $\partial_Y \phi(t, \mathcal{G}_\theta(U)) = \nabla_Y \phi(\mathcal{G}_\theta(U)), \partial_t \phi(t, \mathcal{G}_\theta(U)) = 0$.

The proof can be found in the Appendix A.3. Based on the affine property of $\dot{U}(t)$ in Equation (11), we formulate the following quadratic programming (QP) problem with neural BCBF $\phi$ and neural operator $\mathcal{G}_\theta$ as a safety filter for $\dot{U}_{\text{nominal}}(t), \forall t \in \mathcal{T}$,

$$\dot{U}_{\text{safe}}(t) = \underset{\dot{U} \in \mathbb{R}}{\arg \min} \|\dot{U} - \dot{U}_{\text{nominal}}(t)\| \quad (12)$$

$$s.t. \ \partial_Y \phi(t, Y) \left( \Lambda_\theta(t)\dot{U} + \mu_\theta(t) \right) + \partial_t \phi(t, Y) + \alpha \phi(t, Y) + C_{\alpha,T} \phi(0, U_{\text{nominal}}(0)) \leq 0, \quad (13)$$

where $C_{\alpha,T} = \frac{\alpha}{e^{\alpha T}-1}$ and $\Lambda_\theta(t), \mu_\theta(t)$ can be found in Equation (11). Based on $\dot{U}_{\text{safe}}(t)$ at each step $t$, we find the boundary control input $U_{\text{safe}}(t)$ based on Equation (14) below so that the predicted boundary output $Y_{\text{predict}}(t)$ can be found by the neural operator. Therefore, the next QP update can be solved for $\dot{U}_{\text{safe}}$ at the next time by Equation (12). Note that we let $\dot{U}_{\text{safe}} = \dot{U}_{\text{nominal}}$ for the unfiltered time steps during the QP iteration. The discrete-time implementation of the safety filtering procedure is shown in Algorithm 1.

$$U_{\text{safe}}(t) = \int_0^t \dot{U}(\tau)d\tau + U_{\text{nominal}}(0), \dot{U}(\tau) = \begin{cases} \dot{U}_{\text{safe}}(\tau), \text{if } \|\dot{U}_{\text{safe}}(\tau) - \dot{U}_{\text{nominal}}(\tau)\| \leq \eta, \\ \dot{U}_{\text{nominal}}(\tau), \text{ otherwise.} \end{cases} \quad (14)$$

We remark that iterative filtering with the prediction of $Y(t)$ at each step aims to avoid large approximation errors in Equation (11) in the discrete-time setting compared to one-time filtering for the whole trajectory. Besides, as the computation of QP is not yet real-time, it is not yet ready to interact with the real PDE dynamics. we adopt the predicted $Y(t)$ from the neural operator after each filtering step instead of real PDE dynamics. To handle the model mismatch issue between neural operator modeling and real underlying PDE dynamics, filtering threshold $\eta > 0$ is introduced as a workaround and we leave the study of model mismatch of PDE dynamics as future work. Specifically, the safety filter is disabled when $\eta = 0$. The larger $\eta$ is, the more boundary feasibility within the safe set will be achieved, showing a trade-off between stabilization and constraint satisfaction.

---

**Algorithm 1** Safety Filtering Procedure for Discrete-time Implementation

---

1: **Input:** Nominal control input $U_{1:M}^{\text{nominal}}$, neural operator $\mathcal{G}$, neural BCBF $\phi$, filter threshold $\eta$
2: **Output:** Filtered safe control input $U_{1:M}^{\text{safe}}$
3: Initialize $\Delta U_{1:M}^{\text{safe}} = \Delta U_{1:M}^{\text{nominal}} \leftarrow U_{1:M}^{\text{nominal}} - U_{0:M-1}^{\text{nominal}}, Y_{1:M}^{\text{predict}} \leftarrow \mathcal{G}(U_{1:M}^{\text{nominal}})$
4: **for** $m = 1 : M$ **do**
5:     Find $\Delta U_m^{\text{safe}}$ through QP in Equation (12) based on $\Delta U_m^{\text{nominal}}, Y_{1:M}^{\text{predict}}, \mathcal{G}, \phi, U_0^{\text{nominal}}$
6:     Find $U_{1:M}^{\text{safe}}$ based on Equation (14) with $\Delta U_{1:M}^{\text{safe}}$ and filter threshold $\eta$
7:     Update $Y_{1:M}^{\text{predict}} \leftarrow \mathcal{G}(U_{1:M}^{\text{safe}})$
8: **end for**
9: **return** $U_{1:M}^{\text{safe}}$

---

## 4 EXPERIMENT

In this section, we aim to answer the following two questions: How does the proposed safety filtering perform compared to the vanilla model-free controllers in unknown PDE dynamics? How do filtering thresholds, different convergence types and neural operator modeling influence the performance of the proposed safety filtering? We answer the first question in Section 4.2 and the second one in Section 4.3, following the experimental setup of PDE dynamics, controllers, and evaluation metrics.

### 4.1 EXPERIMENTAL SETUP

**Environments and model-free controllers.** We adopt the challenging PDE boundary control environments as well as the model-free reinforcement learning (RL) controllers from Bhan et al. (2024) to conduct our experiment. More specifically, the three environments include the unstable 1D hyperbolic (transport) equation, 1D parabolic (reaction-diffusion) equation and 2D nonlinear Navier-Stokes equation, where the last one is for tracking task and others are for stabilization task. Since our setting in Problem 2.1 does not have prior to the PDE equations, we choose the model-free RL controllers, PPO Schulman et al. (2017) and SAC Haarnoja et al. (2018), from Bhan et al. (2024) as the baselines in each environment for fair comparisons. The boundary control inputs are consistent with Bhan et al. (2024). For 1D environments, the boundary input is $U(t) = u(1, t)$ while the boundary output for the hyperbolic PDE is $Y(t) = u(0, t)$ and the boundary output for the parabolic PDE $Y(t) = u(0.5, t)$ since $u(0, t) \equiv 0$. For the 2D environment, the boundary input is the x-axis consistent boundary condition, i.e., $u(x, 1, t) \equiv U(x), v(x, 1, t) \equiv 0, \forall x \in [0, 1]$. The boundary output is $Y(t) = u(0.5, 0.95, t), v(x, 0.95, t) \equiv 0, \forall x \in [0, 1]$, which has the maximum speed except for control input and can be viewed as an indicator for tracking performance. Note that we focus on the boundary output that only depends on time in high-dimensional cases. We specify one-sided safe sets $\mathcal{S}_0 = \{Y : AY < b\}$ for stabilization tasks and two-sided safe sets $\mathcal{S}_0 = \{Y : |Y - Y_{gt}| < b\}$ for tracking tasks. With the pre-trained RL models, we collect 50k pairs of boundary input $U(t)$ and output $Y(t)$ trajectory with label annotations based on user-specified safe sets $\mathcal{S}_0$. The temporal resolution of collected trajectories is consistent with the control frequency of each environment in Bhan et al. (2024), i.e. 50 steps in 5s for hyperbolic PDE, 1000 steps in 1s for parabolic PDE and 200 steps in 0.2s for Navier-Stokes PDE. More details can be found in the Appendix B.

**Model training and evaluation metrics.** With the collected dataset from RL models, we train the neural operators and neural BCBFs according to Equation (7) through empirical risk minimization. We adopt the Fourier neural operator (FNO) Li et al. (2020a) as the default neural operator model and train it with Markov neural operator (MNO) Li et al. (2022) using the default hyper-parameters. For the neural BCBF training, following Zhang et al. (2023); Hu et al. (2024), we use a 4-layer feedforward neural network with ReLU activations to parameterize BCBFs and incorporate Equation (4) and Equation (7) with default $\alpha = 10^{-5}$ into the regular model training pipeline Zhao et al. (2020); Dawson et al. (2022) to train both time-independent BCBF $\phi(Y)$ and time-dependent BCBF $\phi(t, Y)$. More details can be found in Appendix B. With the well-trained neural operator and neural BCBF, we solve the QP of Equation (12) though CPLEX IBM and the final control trajectory is found through Equation (14) with threshold $\eta = 2$ as default, mitigating the discrepancy between the PDE environment and the neural operator. For the evaluation of safety filtering for RL controllers, we keep the original RL rewards from Bhan et al. (2024) as a metric to show if the performance

Table 1: Results comparison under 1D hyperbolic transport equation among 100 episodes. The boundary feasibility constraint is $Y < 1$ for PPO and $Y < 0$ for SAC models.

| Models w/o and w. filtering | Reward (mean±std) (starting at ∼-300) | Feasible Rate (100 episodes) | Average Feasible Steps ( 50 control steps) |
|---|---|---|---|
| PPO Bhan et al. (2024) | 157.9±37.5 | 0.63 | 7.6 |
| PPO with filtering of $\phi(Y)$ | 162.3±44.5 | 0.63 | 8.3 |
| PPO with filtering of $\phi(t, Y)$ | **165.0**±43.7 | **0.71** | **9.8** |
| SAC Bhan et al. (2024) | **106.2**±98.7 | 0.78 | 12.4 |
| SAC with filtering of $\phi(Y)$ | 103.3±98.4 | 0.57 | **15.7** |
| SAC with filtering of $\phi(t, Y)$ | 103.4±96.4 | **0.85** | 13.9 |

Table 2: Results comparison under 1D parabolic reaction-diffusion equation among 100 episodes. The boundary feasibility constraint is $Y < 0.6$ for PPO and $Y > -0.26$ for SAC models.

| Models w/o and w. filtering | Reward (mean±std) (starting at ∼0) | Feasible Rate (100 episodes) | Average Feasible Steps ( 1000 control steps) |
|---|---|---|---|
| PPO Bhan et al. (2024) | 164.5±20.7 | 0.60 | 155.0 |
| PPO with filtering of $\phi(Y)$ | 162.9±19.6 | 0.46 | **519.4** |
| PPO with filtering of $\phi(t, Y)$ | **168.2**±23.5 | **0.81** | 507.0 |
| SAC Bhan et al. (2024) | 156.5±6.2 | 0.72 | 118.4 |
| SAC with filtering of $\phi(Y)$ | **157.9**±6.9 | **0.92** | **543.2** |
| SAC with filtering of $\phi(t, Y)$ | 157.5±6.8 | 0.87 | 449.8 |

is compromised by the enhancement of safety constraints. Besides, we introduce two new metrics regarding boundary feasibility, *Feasible Rate* and *Average Feasible Steps*. *Feasible Rate* is the ratio of trajectories that boundary feasibility in Definition 2.1 is achieved, i.e., the boundary output falls into the safe set and will not go out of it by the end of a single trajectory with finite steps. *Average Feasible Steps* is the mean steps among boundary feasible trajectories in which the boundary output is consistently kept in the safe set until the end of the trajectory, characterizing how long the boundary feasibility is achieved and maintained.

## 4.2 RESULTS COMPARISON

**1D Hyperbolic (transport) PDE.** Table 1 shows the results from different model-free RL controllers without and with safety filtering under time-independent BCBF $\phi(Y)$ and time-dependent BCBF $\phi(t, Y)$. Both PPO and SAC with filtering outperform the vanilla PPO and SAC in feasible rate and average feasible steps, validating the effectiveness of the proposed safety filtering method. Specifically, PPO with filtering of $\phi(t, Y)$ presents the highest feasible rate and largest average feasible steps, showing that time-dependent BCBF can distinguish the feasibility of the PDE boundary state more effectively by explicitly taking time as an input compared to the time-independent one. Regarding the reward comparison, the safety filtering of the PPO model with $\phi(Y)$ and $\phi(t, Y)$ results in a higher reward than the PPO baseline. This is because the safety constraint $Y < 1$ can be aligned with the task of stabilization $Y \to 0$, i.e., a safer trajectory can come with a higher reward. However, safety filtering for SAC models compromises the stabilization performance with lower reward, due to the conflicted constraint satisfaction $Y < 0$ and stabilization goal to 0.

**1D Parabolic (reaction-diffusion) PDE.** As shown in Table 2, since the boundary feasibility constraint $Y < 0.6$ or $Y > -0.26$ is not conflict with the stabilization goal $Y \to 0$, the safety filtering can also boost the reward metric compared to the vanilla PPO and SAC. Feasible rate for PPO with $\phi(t, Y)$ filtering is the highest but its average feasible step is lower than $\phi(Y)$ filtering, because time-independent BCBF $\phi(Y)$ tends to have divergent performance with more non-feasible trajectories and more feasible steps for feasible trajectories. However, with SAC models, time-independent BCBF $\phi(Y)$ works the best in all metrics because of the lower variance of the baseline SAC model and consistent trajectory pattern, making boundary feasibility less related to time and easier to learn without explicitly incorporating $t$ into BCBF. In this case, it is more challenging to learn $\phi(t, Y)$ with larger data complexity, resulting in sightly worse performance than $\phi(Y)$.

**2D Navier-Stokes PDE.** From Table 3, we can see that compared to the vanilla PPO and SAC, the results of safety filtering with time-dependent BCBF $\phi(t, Y)$ are better in the metrics of feasible

Table 3: Results comparison under nonlinear 2D Navier–Stokes equation among 100 episodes. The boundary feasibility constraint is $|Y - Y_{gt}| < 0.145$ for PPO and SAC models.

| Models w/o and w. filtering | Reward (mean±std) (starting at ∼-100) | Feasible Rate (100 episodes) | Average Feasible Steps ( 200 control steps) |
|---|---|---|---|
| PPO Bhan et al. (2024) | **-5.37**±0.01 | 0.86 | 2.0 |
| PPO with filtering of $\phi(Y)$ | **-5.37**±0.01 | 0.86 | 2.2 |
| PPO with filtering of $\phi(t, Y)$ | -5.72±0.17 | **0.99** | **32.0** |
| SAC Bhan et al. (2024) | **-18.05**±1.13 | 0.80 | 17.5 |
| SAC with filtering of $\phi(Y)$ | **-18.05**±1.14 | 0.79 | 17.8 |
| SAC with filtering of $\phi(t, Y)$ | -18.36±1.25 | **0.85** | **21.3** |

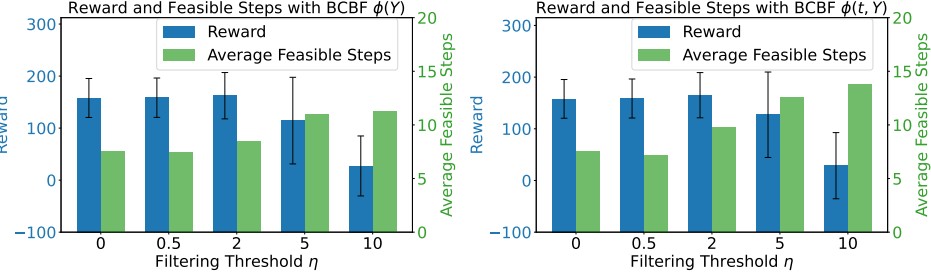

Figure 2: The reward and feasible rate under different filtering threshold $\eta$ in Equation (14) with BCBF $\phi(Y)$ (left) and $\phi(t, Y)$ (right) for PPO model in hyperbolic equation. Note that $\eta = 0$ indicates the vanilla PPO model without safety filtering.

rate and average feasible steps, while the rewards after filtering get compromised. The reason lies in that the relaxed safe set $|Y - Y_{gt}| < 0.145$ only enforces constraint over a specific high-speed boundary instead of the whole 2D plane, which is used to calculate the reward. Regarding the filtering with different types of BCBFs, $\phi(Y)$ has limited improvement over the baseline models but time-dependent one $\phi(t, Y)$ shows significant superiority over feasible rate and steps, especially for the PPO model. This implies that BCBF $\phi(t, Y)$ does better in capturing complicated feasibility over the marginally observed PDE state that only depends on time in high-dimensional cases.

### 4.3 ABLATION STUDY AND DISCUSSION

**Influence of filtering threshold.** In this section, we investigate the influence of filtering threshold $\eta$ in Equation (14) to show the trade-off between general performance and boundary feasibility. From Figure 2, it can be seen that as the threshold goes up, the reward first slightly increases and then drops significantly, showing that the strong safety filtering may hurt the stability of the PPO controller due to the model mismatch between direct boundary mapping with the neural operator and underlying PDE dynamics. Besides, with a larger filtering threshold $\eta$, the average feasible steps become larger as the safety filtering becomes stronger, especially for time-dependent BCBF $\phi(t, Y)$, guaranteeing constraint satisfaction over boundary output. With small $\eta$, the average feasible steps may be less than the one without filtering because of more feasible trajectories with last-step feasibility. More details can be found in Appendix B.2.

**Comparison of asymptotic and finite-time boundary feasibility.** In Table 4, we show the comparison of safety filtering with BCBF $\phi(t, Y)$ for 1D hyperbolic equation for asymptotic and finite-time boundary feasibility. Asymptotic boundary feasibility is with the neural BCBF trained and tested with $C_{\alpha,T} = \lim_{T \to \infty} \frac{\alpha}{e^{\alpha T} - 1} = 0$ while finite-time boundary feasibility is with $C_{\alpha,T} = 0.02$ using $T = 50$. It can be seen that BCBF with finite-time feasibility has a better feasible rate, especially the SAC model, as asymptotic feasibility is weaker than finite-time feasibility and takes longer steps to converge. However, for general performance of reward, since asymptotic feasibility causes weaker filtering effects, the reward tends to be closer to the vanilla reward without filtering in Table 1 compared to finite-time feasibility, which is validated in Table 4.

**Boundary mapping with different neural operators.** Here we compare two neural operators, FNO Li et al. (2020a) and MNO Li et al. (2022), for learning the boundary mapping from control input $U(t)$ to output $Y(t)$ for 1D hyperbolic equation in Table 5. With the same time-dependent

Table 4: Results of filtering with BCBF $\phi(t, Y)$ for 1D hyperbolic equation for asymptotic $C_{\alpha,T} = \lim_{T \to \infty} \frac{\alpha}{e^{\alpha T} - 1} = 0$ and finite-time $C_{\alpha,T} = \frac{\alpha}{e^{\alpha T} - 1} = 0.02$ at $T = 50, \alpha = 10^{-5}$.

| Different neural operators | Reward (mean±std) (starting at ∼-300) | Feasible Rate (100 episodes) | Average Feasible Steps ( 50 control steps) |
|---|---|---|---|
| PPO for asymptotic feasibility | 163.8±40.6 | 0.70 | 8.1 |
| PPO for finite-time feasibility | **165.0**±43.7 | **0.71** | **9.8** |
| SAC for asymptotic feasibility | **104.6**±98.6 | 0.56 | **14.7** |
| SAC for finite-time feasibility | 103.4±96.4 | **0.85** | 13.9 |

Table 5: Filtering with BCBF $\phi(t, Y)$ under different neural operators for 1D hyperbolic equation.

| Different neural operators | Reward (mean±std) (starting at ∼-300) | Feasible Rate (100 episodes) | Average Feasible Steps ( 50 control steps) |
|---|---|---|---|
| PPO w. MNO Li et al. (2022) | 163.8±47.2 | **0.78** | 9.0 |
| PPO w. FNO Li et al. (2020a) | **165.0**±43.7 | 0.71 | **9.8** |
| SAC w. MNO Li et al. (2022) | 103.3±96.4 | 0.84 | **14.7** |
| SAC w. FNO Li et al. (2020a) | **103.4**±96.4 | **0.85** | 13.9 |

BCBF $\phi(t, Y)$, the safety filtering with FNO presents higher rewards under both PPO and SAC base models, showing that FNO is more suitable for learning low-resolution trajectories with 50 sampled points. Besides, MNO shows better feasible rate and average feasible steps performance, especially with SAC as the base model, since the MNO model has a larger model complexity.

**Qualitative visualization.**   In this section, we visualize and compare multiple trajectories under 1D hyperbolic equation using PPO controller without and with safety filtering of $\phi(t, Y)$, as shown in Figure 3. We can see that for each trajectory, the state value $u(x, t)$ after filtering is lower than that before filtering. More specifically, as time goes by, the filtered control input $U(t)_{\text{safe}}$ in blue dashed lines deviates more away from nominal control input $U(t)_{\text{nominal}}$ in red dashed lines, causing the filtered boundary output $Y(t)_{\text{safe}}$ in blue solid lines to satisfy the constraint $Y(t) < 1$ compared to the nominal boundary output $Y(t)_{\text{nominal}}$ in red solid lines.

## 5   RELATED WORK

**Control for PDE Dynamics.**   PDE control problems can be in-domain control Botteghi & Fasel (2024); Zhang et al. (2024b) or boundary control Krstic & Smyshlyaev (2008b); Smyshlyaev & Krstic (2010), where the latter is more commonly-seen setting in the real world. As it has been studied for over a decade, backstepping has become a dominant approach for boundary control with known PDE dynamics Krstic & Smyshlyaev (2008a); Smyshlyaev & Krstic (2004). Recently, learning-based controllers have gotten rid of the requirement of analytical form of unstable PDE dynamics and become a promising solution to the PDE control problems Botteghi & Fasel (2024); Zhang et al. (2024b); Krstic et al. (2024); Qi et al. (2023); Mowlavi & Nabi (2023). However, regarding the safety of constraint satisfaction in the PDE dynamics, current backstepping-based safe PDE control methods (Krstic & Bement, 2006; Li & Krstic, 2020; Koga & Krstic, 2023; Wang & Krstic, 2023) still assume the non-stable PDE dynamics is known. Therefore, we focus on data-driven methods for PDE safe control without any prior knowledge of PDE dynamics.

**Safe Control with Neural Certificate**   For the control of the ODE dynamical system, there is rich literature regarding learning-based controllers with safety guarantees or certificates Boffi et al. (2021); Dawson et al. (2023); Xiao et al. (2023); Lindemann et al. (2021); Chang et al. (2019); Mazouz et al. (2022). Neural networks have been used to parameterize the CBFs under complex dynamics with bounded control inputs Liu et al. (2022); So et al. (2023); Zinage et al. (2023); Dawson et al. (2022); Dai et al. (2022), which result in forward invariance of the user-specified safe set to guarantee the safety with neural certificate for learning-based controllers Choi et al. (2021); Wei et al. (2022); Agrawal & Panagou (2021); Xiao et al. (2022); Hsu et al. (2023), i.e. once the states enter the safe set, they will never go out. However, forward invariance may not hold in the PDE boundary control setting with commonly-seen highly oscillating trajectories. For example, highly-oscillating trajectories may go out of the safe set during the early oscillation and break the forward invariance defined by conventional ODE CBFs Liu & Tomizuka (2014); Ames et al. (2014),

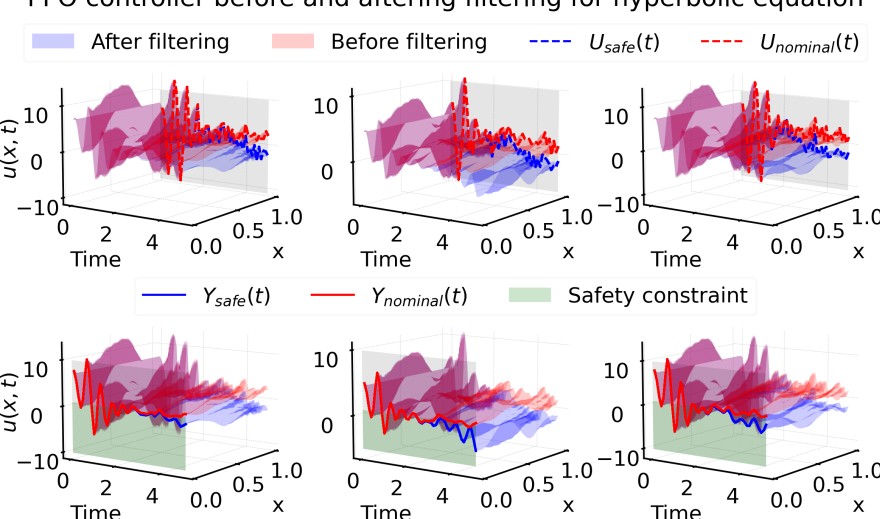

Figure 3: Visualization of three state trajectories $u(x, t)$ (left, mid, right) for hyperbolic equation under PPO controller with and without safety filtering. Boundary control inputs $U(t)$ are in dashed lines and boundary output $Y(t)$ are in solid lines. The boundary constraint $Y(t) < 1$ is in green.

but they could still converge to the constraint satisfaction by the end of time. Therefore, we focus on boundary feasibility, a new notion introduced in this paper. Approach-wise, the CBF-QP for ODE dynamics Liu & Tomizuka (2014); Lindemann & Dimarogonas (2018); Xiao et al. (2021); Garg & Panagou (2021b) does not apply. That is because PDE boundary control does not have Markov property at each control step, due to the infinite-dimensional unobserved non-boundary states. We adopt a neural operator to model the trajectory-to-trajectory mapping and control the change of input boundary through a novel QP formulation.

**Neural Operator Learning for PDEs.** Neural operator learning has become as a powerful tool for solving PDEs by learning mappings between function spaces rather than pointwise approximations Kovachki et al. (2023); Brunton & Kutz (2024). Recent research has demonstrated the utility of neural operators in multiple science and engineering fields like fluid dynamics, weather forecasting, and robotics Kochkov et al. (2021); Pathak et al. (2022); Heiden et al. (2021); Raissi et al. (2019). There exist multiple architectures for neural operators based on different mathematical properties of data. Lu et al. (2021) introduces DeepONet with a branch and a trunk network, and NOMAD Seidman et al. (2022) adopts nonlinear decoder map to learn submanifolds in function spaces, while Green's function-inspired neural operators Li et al. (2020a;b;c; 2022; 2024) adopt linear integral kernel representation with various kernel implementations. However, for the PDE boundary control problem, current works Bhan et al. (2023); Krstic et al. (2024) only adopt neural operators to learn the integral kernel in backstepping, which does not release the full potential of neural operator for characterizing and controlling unknown dynamics. The proposed work is the first to leverage neural operators to learn the direct mapping from control input to boundary output as a transfer function.

## 6 CONCLUSION AND FUTURE WORK

In this work, we introduce a novel safe PDE boundary control framework using safety filtering with neural certification. First, BCBF and neural operator are learned from collected PDE boundary input and output trajectories within a given safe set. Then boundary feasibility is guaranteed by filtering the unsafe boundary conditions using the BCBF. we show that the change in the BCBF depends linearly on the change in input boundary, hence the filtering can be done by solving a quadratic programming problem. Experiments on three challenging PDE control environments validate the effectiveness of the proposed method in terms of both general performance and constraint satisfaction. One limitation of the work is that our work does not consider complicated boundary constraint settings and safe sets. Model mismatch between underlying PDE and neural operator is also an important but unexplored topic, which is marked as future work.

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

# A PROOFS

## A.1 PRELIMINARY

**Definition A.1** (Boundary Feasibility for Trajectory-wise Finite-time Constraint Satisfaction). With state $u(x,t)$ subjected to closed-loop PDE dynamics in Equation (1) with the boundary control input $U(t)$, the boundary control output $Y(t)$ is defined to be feasible over $\mathcal{T}$ within the given user-specified safe set $\mathcal{S}_0 \in \mathcal{S}$ if the following holds,

$$\exists t_0 \in \mathcal{T}, \forall t_0 \le t \le T, Y(t) := u(\mathbf{0}, t) \in \mathcal{S}_0, \text{ where } u(\mathbf{1}, t) = U(t), u(x, 0) \equiv U(0). \tag{15}$$

**Definition A.2** (Neural operator for input-output boundary mapping, reformulated from Section 3.2). Neural operator $\mathcal{G}_\theta : \{U : \mathcal{T} \to \mathcal{S}\} \mapsto \{Y : \mathcal{T} \to \mathcal{S}\}$ can be formalized as

$$Y(t) = \mathcal{G}_\theta(U)(t) = Q(v_L(t)), v_0(t) = P(U(t)), \text{ where each layer } v_l(t) \text{ is} \tag{16}$$

$$v_{l+1}(t) = \mathcal{I}_l(v_l)(t) = \sigma_{l+1}\left(W_l v_l(t) + \int_{\mathcal{T}} \kappa^{(l)}(t, s) v_l(s) ds + b_l(t)\right), l = 0, 1, \ldots, L - 1 \tag{17}$$

where $\sigma_{l+1} : \mathbb{R}^{d_{v_{l+1}}} \to \mathbb{R}^{d_{v_{l+1}}}$ is the activation function, $W_l \in \mathbb{R}^{d_{v_{l+1}} \times d_{v_l}}$ is the local linear operator, $P \in \mathbb{R}^{v_0 \times \dim(\mathcal{S})}$ and $Q \in \mathbb{R}^{\dim(\mathcal{S}) \times v_L}$ are lifting and projection matrix, $\kappa^{(l)} \in C(\mathcal{T} \times \mathcal{T}; \mathbb{R}^{d_{v_{l+1}} \times d_{v_l}})$ is the kernel function for integration, and $b_l \in C(\mathcal{T}; \mathbb{R}^{d_{v_{l+1}}})$ is the bias function. And $P, Q, W_l, \kappa^{(l)}, b_l, l = 0, 1, \ldots, L - 1$ are parameterized with neural networks $\theta$.

## A.2 PROOF OF THEOREM 3.1

**Theorem A.1** (Boundary Feasibility with Boundary Control Barrier Function). For the state $u(x,t)$ from the closed-loop PDE dynamics with boundary control input $U(t) = u(\mathbf{1}, t), u(x, 0) \equiv U_0$, the boundary feasibility of boundary output $Y(t) = u(\mathbf{0}, t)$ over $\mathcal{T} = [0, T]$ within user-specified safe set $\mathcal{S}_0$ is guaranteed with neural BCBF $\phi(t, Y)$ if the following holds $\forall t \in \mathcal{T}$

$$\left(\mathcal{S}_{\phi,t} := \{Y \mid \phi(t, Y) \le 0\} \subseteq \mathcal{S}_0\right) \bigwedge \left(\partial_Y \phi \cdot \frac{dY}{dt} + \partial_t \phi + \alpha \phi(t, Y) + C_{\alpha,T}\phi(0, U_0) \le 0\right) \tag{18}$$

where $C_{\alpha,T} := \frac{\alpha}{e^{\alpha T} - 1} > 0$ is a constant for finite-time convergence. Similarly, the boundary feasibility with neural BCBF $\phi(Y)$ holds if Equation (3) holds by letting $\partial_Y \phi = \nabla_Y \phi, \partial_t \phi = 0$.

*Proof.* To show the boundary feasibility of the boundary output of $Y(t)$ within user-specified safe set $\mathcal{S}_0$, by Definition A.1, we need to show

$$\exists t_0 \in [0, T], s.t. \forall t \in [t_0, T], Y(t) \in \mathcal{S}_0. \tag{19}$$

With the sublevel set $\mathcal{S}_{\phi,t}$ being the subset of $\mathcal{S}_0$, i.e., $\mathcal{S}_{\phi,t} := \{Y \mid \phi(t, Y) \le 0\} \subseteq \mathcal{S}_0$, it is sufficient to prove

$$\exists t_0 \in [0, T], s.t. \forall t \in [t_0, T], \phi(t, Y(t)) \le 0. \tag{20}$$

Now denote $\psi(t) := \phi(t, Y(t))$, by initial constant boundary condition $Y(0) = u(\mathbf{0}, 0) = u(\mathbf{1}, 0) = U_0$, we have the following equivalent inequalities hold,

$$\partial_Y \phi \cdot \frac{dY}{dt} + \partial_t \phi + \alpha \phi(t, Y) + C_{\alpha,T}\phi(0, Y(0)) \le 0 \tag{21}$$

$$\Longleftrightarrow \frac{d\phi(t, Y(t))}{dt} + \alpha\phi(t, Y) + C_{\alpha,T}\phi(0, Y(0)) \le 0 \tag{22}$$

$$\Longleftrightarrow \frac{d\psi(t)}{dt} + \alpha\psi(t) + C_{\alpha,T}\psi(0) \le 0 \tag{23}$$

$$\Longleftrightarrow e^{\alpha t}\frac{d\psi(t)}{dt} + e^{\alpha t}\alpha\psi(t) + e^{\alpha t}C_{\alpha,T}\psi(0) \le 0, \forall t \in [0, T] \tag{24}$$

$$\Longleftrightarrow \frac{d(e^{\alpha t}\psi(t) + \frac{C_{\alpha,T}\psi(0)}{\alpha}e^{\alpha t})}{dt} \le 0 \tag{25}$$

So we have the function $e^{\alpha t}\psi(t) + \frac{C_{\alpha,T}\psi(0)}{\alpha}e^{\alpha t}$ be non-increasing over $t \in [0, T]$. By $T > 0$, we have

$$[e^{\alpha t}\psi(t) + \frac{C_{\alpha,T}\psi(0)}{\alpha}e^{\alpha t}]|_{t=T} < [e^{\alpha t}\psi(t) + \frac{C_{\alpha,T}\psi(0)}{\alpha}e^{\alpha t}]|_{t=0} \tag{26}$$

$$\iff e^{\alpha T}\psi(T) + \frac{e^{\alpha T}}{e^{\alpha T} - 1}\psi(0) < \psi(0) + \frac{1}{e^{\alpha T} - 1}\psi(0) \tag{27}$$

$$\iff e^{\alpha T}\psi(T) < 0 \tag{28}$$

$$\iff \psi(T) < 0 \tag{29}$$

$$\iff \phi(T, Y(T)) < 0 \tag{30}$$

So at least at $t_0 = T$, $\phi(t_0, Y(t_0)) < 0$, which proves Equation (20) holds and the original theorem has been proved. Furthermore, let us look at the boundary feasible steps. Since $e^{\alpha t}\psi(t) + \frac{C_{\alpha,T}\psi(0)}{\alpha}e^{\alpha t} = e^{\alpha t}(\psi(t) + \frac{C_{\alpha,T}\psi(0)}{\alpha})$ is non-increasing, with the strictly increasing and positive $e^{\alpha t}$, it is easy to find function $\psi(t) + \frac{C_{\alpha,T}\psi(0)}{\alpha}$ being non-increasing, i.e. $\psi(t)$ is non-increasing. Therefore, if $U_0 \le 0$, $\phi(t, Y(t)) < \phi(0, Y(0)) = U_0 < 0, \forall t \in [0, T]$. If $U_0 > 0$, since MLP-ReLU parameterized neural BCBF $\phi$ and boundary control output $Y$ are continuous, by mean value theorem, we have

$$\phi(0, Y(0)) > 0, \phi(T, Y(T)) < 0 \Rightarrow \exists t_0 \in [0, T], \phi(t_0, Y(t_0)) = 0. \tag{31}$$

Since $\psi(t) = \phi(t, Y(t))$ is non-increasing, we have

$$\exists t_0 \in [0, T], s.t. \forall t \in [t_0, T], \phi(t, Y(t)) \le 0, \tag{32}$$

which concludes the proof. $\qquad\square$

### A.3 PROOF OF THEOREM 3.2

**Theorem A.2** (Boundary Feasibility with Neural Operator). Assuming the neural operator $\mathcal{G}_\theta$ as an exact map from boundary input $U(t)$ to output $Y(t)$ for an unknown closed-loop PDE dynamics without model mismatch, the boundary control input $U(t)$ is guaranteed to induce boundary feasibility of output $Y(t)$ over $\mathcal{T} = [0, T]$ within the sublevel set of neural BCBF $\phi$ if $U(t)$ satisfies

$$\partial_Y \phi(t, \mathcal{G}_\theta(U))\frac{d\mathcal{G}_\theta(U)(t)}{dt} + \partial_t \phi(t, \mathcal{G}_\theta(U)) + \alpha\phi(t, \mathcal{G}_\theta(U)) + C_{\alpha,T}\phi(0, U(0)) \le 0, \forall t \in \mathcal{T} \tag{33}$$

where $C_{\alpha,T} = \frac{\alpha}{e^{\alpha T} - 1}$, and $\frac{d\mathcal{G}_\theta(U)(t)}{dt}$ can be found below with $\prod_1^0(\cdot) := 1$,

$$\frac{d\mathcal{G}_\theta(U)(t)}{dt} = \nabla Q^\top \prod_{l=0}^{L-1}\left(\text{Diag}(\sigma'_{L-l})W_{L-1-l}\right)\nabla P^\top \frac{dU(t)}{dt} + \nabla Q^\top \text{Diag}(\sigma'_L)\sum_{i=0}^{L-1}\left([\prod_{j=1}^i W_{L-j}\right.$$

$$\left.\text{Diag}(\sigma'_{L-j})]\left(\int_{\mathcal{T}}\frac{\partial\kappa^{(L-1-i)}(t,s)}{\partial t}v_{L-1-i}(s)ds + \frac{db_{L-1-i}(t)}{dt}\right)\right) = \Lambda_\theta(t)\dot{U}(t) + \mu_\theta(t) \tag{34}$$

*Proof.* To show the boundary feasibility over sublevel set of $\phi$ hold, we first want to show Equation (34) holds. According to Definition A.2, we first rewrite the neural operator as

$$Y(t) = \mathcal{G}_\theta(U)(t) = Q(v_L(t)), v_0(t) = P(U(t)), \text{ where each layer } v_l(t) \text{ is}$$

$$v_{l+1}(t) = \mathcal{I}_l(v_l)(t) = \sigma_{l+1}\left(W_l v_l(t) + \int_{\mathcal{T}}\kappa^{(l)}(t,s)v_l(s)ds + b_l(t)\right), l = 0, 1, \ldots, L-1 \tag{35}$$

where $P, Q, W_l, \kappa^{(l)}, b_l, l = 0, 1, \ldots, L-1$ are neural networks, kernel function $\kappa^{(l)}$, activation function $\sigma_l$ and bias function $b_l$ are first-order differential. Since the operator shares the same input function domain and output function domain over $t \in \mathbb{R}^+$, applying chain rule to Equation (35), we can find the derivative with respect to $t$ for each layer as,

$$\frac{dY(t)}{dt} = \nabla Q^\top\frac{dv_L(t)}{dt}, \frac{v_0(t)}{dt} = \nabla P^\top\frac{dU(t)}{dt}, \text{ for each derivative }\frac{dv_{l+1}(t)}{dt} l = L-1, \ldots, 0,$$

$$\tag{36}$$

$$\frac{dv_{l+1}(t)}{dt} = \mathcal{J}_l(\frac{dv_l}{dt})(t) = \text{Diag}(\sigma'_{l+1})\left(W_l\frac{dv_l(t)}{dt} + \int_{\mathcal{T}}\frac{\partial\kappa^{(l)}(t,s)}{\partial t}v_l(s)ds + \frac{db_l(t)}{dt}\right) \tag{37}$$

Now put Equation (37) into Equation (36) recursively, we have

$$\frac{d\mathcal{G}(U)(t)}{dt} = \nabla Q^\top \frac{dv_L(t)}{dt} \tag{38}$$

$$=\nabla Q^\top \mathrm{Diag}(\sigma_L')W_{L-1}\frac{dv_{L-1}(t)}{dt} + \nabla Q^\top \mathrm{Diag}(\sigma_L')\left(\int_{\mathcal{T}}\frac{\partial\kappa^{(L-1)}(t,s)}{\partial t}v_{L-1}(s)ds + \frac{db_{L-1}(t)}{dt}\right) \tag{39}$$

$$=\nabla Q^\top \mathrm{Diag}(\sigma_L')W_{L-1}\mathrm{Diag}(\sigma_{L-1}')W_{L-2}\frac{dv_{L-2}(t)}{dt} + \nabla Q^\top \mathrm{Diag}(\sigma_L')W_{L-1}\cdot\mathrm{Diag}(\sigma_{L-1}')\cdot$$

$$\left(\int_{\mathcal{T}}\frac{\partial\kappa^{(L-2)}(t,s)}{\partial t}v_{L-2}(s)ds + \frac{db_{L-2}(t)}{dt}\right) + \nabla Q^\top \mathrm{Diag}(\sigma_L')(\int_{\mathcal{T}}\frac{\partial\kappa^{(L-1)}(t,s)}{\partial t}v_{L-1}(s)ds$$

$$+ \frac{db_{L-1}(t)}{dt}) \tag{40}$$

$$= \ldots \text{(recursively apply Equation (37))}$$

$$=\nabla Q^\top \mathrm{Diag}(\sigma_L')W_{L-1}\ldots\mathrm{Diag}(\sigma_1')W_0\frac{dv_0(t)}{dt} + \nabla Q^\top \mathrm{Diag}(\sigma_L')W_{L-1}\mathrm{Diag}(\sigma_{L-1}')\cdots W_1$$

$$\mathrm{Diag}(\sigma_1')\left(\int_{\mathcal{T}}\frac{\partial\kappa^{(0)}(t,s)}{\partial t}v_0(s)ds + \frac{db_0(t)}{dt}\right) + \cdots + \nabla Q^\top \mathrm{Diag}(\sigma_L')W_{L-1}\cdot\mathrm{Diag}(\sigma_{L-1}')\cdot$$

$$\left(\int_{\mathcal{T}}\frac{\partial\kappa^{(L-2)}(t,s)}{\partial t}v_{L-2}(s)ds + \frac{db_{L-2}(t)}{dt}\right) + \nabla Q^\top \mathrm{Diag}(\sigma_L')(\int_{\mathcal{T}}\frac{\partial\kappa^{(L-1)}(t,s)}{\partial t}v_{L-1}(s)ds$$

$$\frac{db_{L-1}(t)}{dt}) \tag{41}$$

$$=\nabla Q^\top \prod_{l=0}^{L-1}\left(\mathrm{Diag}(\sigma_{L-l}')W_{L-1-l}\right)\nabla P^\top \frac{dU(t)}{dt} + \nabla Q^\top \mathrm{Diag}(\sigma_L')\sum_{i=0}^{L-1}\left([\prod_{j=1}^{i}W_{L-j}\mathrm{Diag}(\sigma_{L-j}')]\cdot\right.$$

$$\left.\left(\int_{\mathcal{T}}\frac{\partial\kappa^{(L-1-i)}(t,s)}{\partial t}v_{L-1-i}(s)ds + \frac{db_{L-1-i}(t)}{dt}\right)\right) \tag{42}$$

Note that the final expression in Equation (42) is actually linear with respect to $\dot{U}(t)$ and the weight and bias terms only depend on the parameters of the neural operator $\theta$ and the values at time $t$. Denote the linear weight and bias as $\Lambda_\theta(t), \mu_\theta(t)$

$$\Lambda_\theta(t) := \nabla Q^\top \prod_{l=0}^{L-1}\left(\mathrm{Diag}(\sigma_{L-l}')W_{L-1-l}\right)\nabla P^\top, \mu_\theta(t) := \nabla Q^\top \mathrm{Diag}(\sigma_L')\cdot \tag{43}$$

$$\sum_{i=0}^{L-1}\left([\prod_{j=1}^{i}W_{L-j}\mathrm{Diag}(\sigma_{L-j}')]\cdot\left(\int_{\mathcal{T}}\frac{\partial\kappa^{(L-1-i)}(t,s)}{\partial t}v_{L-1-i}(s)ds + \frac{db_{L-1-i}(t)}{dt}\right)\right), \tag{44}$$

then we have

$$\frac{dY(t)}{dt} = \frac{d\mathcal{G}(U)(t)}{dt} = \Lambda_\theta(t)\dot{U}(t) + \mu_\theta(t).$$

Since $Y(t) = \mathcal{G}(U)(t)$, Equation (33) is equivalent to

$$\partial_Y\phi \cdot \frac{dY}{dt} + \partial_t\phi + \alpha\phi(t,Y) + C_{\alpha,T}\phi(0,U(0)) \le 0.$$

Similar to the proof of Theorem A.1, we have

$$\exists t_0 \in [0,T], s.t.\forall t \in [t_0,T], \phi(t,Y(t)) \le 0, \tag{45}$$

which concludes the proof of boundary feasibility over the sublevel set of $\phi$. $\qquad\square$

Table 6: Comparison of before QP and after QP filtering with different thresholds using $\phi(Y)$ and $\phi(t, Y)$ for PPO model under hyperbolic equation.

| Filtering with $\phi(Y)$ | Reward (mean±std) | Feasible Rate | Average Feasible Steps |
|---|---|---|---|
| Before QP (baseline) | 157.90±37.46 | 0.63 | 7.56 |
| After QP with threshold 0.5 | 158.45±37.82 | 0.65 | 7.49 |
| After QP with threshold 2 | 162.26±44.53 | 0.63 | 8.49 |
| After QP with threshold 5 | 114.40±83.25 | 0.67 | 11.01 |
| After QP with threshold 10 | 27.28±57.62 | 0.57 | 11.30 |
| Filtering with $\phi(t, Y)$ | Reward (mean±std) | Feasible Rate | Average Feasible Steps |
| Before QP (baseline) | 157.90±37.46 | 0.63 | 7.56 |
| After QP with threshold 0.5 | 158.60±37.76 | 0.68 | 7.19 |
| After QP with threshold 2 | 165.04±43.73 | 0.71 | 9.80 |
| After QP with threshold 5 | 127.18±82.67 | 0.73 | 12.60 |
| After QP with threshold 10 | 28.61±64.03 | 0.57 | 13.74 |

## B  EXPERIMENT DETAILS

### B.1  EXPERIMENT SETTING

**Data preparation.** We train the RL models PPO and SAC following the default hyper-parameters and unstable PDE settings Bhan et al. (2024) for hyperbolic and parabolic equations, while directly adopting the pre-trained models under default Navier-Stokes equation Bhan et al. (2024). For the data collection in the 1D hyperbolic equation, we evaluate the backstepping-based model Krstic & Smyshlyaev (2008a), PPO and SAC models with random initial conditions $U_0 \in [1, 10]$ and collect 50k pairs of input and output $u(1, t), u(0, t)$ trajectories for each model. Similarly, for the 1D parabolic equation, we evaluate the backstepping-based model Smyshlyaev & Krstic (2004), PPO and SAC models with random initial conditions $U_0 \in [1, 10]$ and collect 50k pairs of input and output $u(1, t), u(0.5, t)$ trajectories for each model. For the Navier-Stokes equation, we evaluate the model-based optimization method Pyta et al. (2015), PPO and SAC models with random initial conditions $u_0 \in [-0.1, 0.1]$ and default tracking ground truth and collect 10k pairs of input and output $u(0.05, 1, t), u(0.5, 0.95, t)$ trajectories for each model. After the data pairs are collected, we annotate the safety label with pre-defined safe constraints based on the original performance of each policy: for the hyperbolic equation, $Y < 1$ for PPO and $Y < 0$ for SAC; for the parabolic equation, $Y < 0.6$ for PPO and $Y > -0.26$ for SAC; for the Navier-Stokes equation, $|Y - Y_{gt}| < 0.145$ for PPO and SAC models. Then we randomly split 90% as a training dataset and leave others as a test set.

**Model training.** To train the neural operator models, we adopt the public package NeuralOperators.jl, using the default gelu-activation model of FNO with channels of $(2, 64, 64, 64, 64, 64, 128, 1)$ and 16 modes, MNO with channels of $(2, 64, 64, 64, 64, 64, 1)$ and 16 modes. All the models are trained for 100 epochs with learning rate $10^{-3}$, $\ell$-2 regularization weight is $10^{-4}$, ADAM optimizer and $\ell$-2 loss. The resolutions and scales of hyperbolic, parabolic, and Navier-Stokes trajectories are 50, 1000, and 200 for 5s, 1s, and 0.2s, respectively. We keep the same setting for different environments and remark that we do not fully exploit the potential for the best performance of neural operators since it is not the main focus of this work. For the neural BCBF training, we directly use the finite difference of $Y(t)$ collected from real PDE dynamics instead of the gradient of the neural operator to avoid noise. Following the implementation of Dawson et al. (2022); Zhang et al. (2023); Hu et al. (2024), we adopt 4-layer MLPs with ReLU with layer dimensions of (16,64,16,1) to model neural BCBFs. The time $t$ is concatenated with $Y(t)$ as input for time-dependent neural BCBF $\phi(t, Y)$ while only $Y(t)$ is input for time-independent neural BCBF $\phi(t, Y)$. To construct the safe

Table 7: Results of filtering with BCBF $\phi(Y)$ under different neural operator modeling for first-order transport equation. The boundary feasibility constraint is $Y < 1$ for PPO and $Y < 0$ for SAC models.

| Filtering with different BCBFs | Reward (mean±std) (starting at ∼-300) | Feasible Rate (100 episodes) | Average Feasible Steps ( 50 control steps) |
|---|---|---|---|
| PPO w. MNO | **162.9**±45.2 | **0.68** | **8.7** |
| PPO w. FNO | 162.3±44.5 | 0.63 | 8.3 |
| SAC w. MNO | 103.2±98.3 | **0.59** | 15.4 |
| SAC w. FNO | **103.3**±98.4 | 0.57 | **15.7** |

set loss in Equation (4), we adopt all the sampled steps along trajectories with unsafe labels while only choosing the "latest" safe sampled steps where boundary feasibility is satisfied in Definition 2.1, i.e. once $Y(t)$ is with safe label, it will never become unsafe in finite time $T$. For the boundary feasibility loss in Equation (7), due to too much data close to 0, we adopt randomly drop close-0 data to balance the output boundary data distribution. Specifically, for the hyperbolic equation, we keep 20% data within [-0.1,0.1] while keeping 20% data within [-0.01,0.01] for the parabolic equation. Following Liu et al. (2022), we adopt regularization loss to avoid the shrinking of the sublevel set during training with a default weight of 1. We train all models with ADAM for 20 epochs with an initial learning rate of 0.01. The learning rate decay rate is 0.2 after each 4 epoch. The code is zipped as the supplementary material.

## B.2 ADDITIONAL RESULTS

**Quantitative results under different filtering thresholds.** As shown in Table 6, we can find the reward and feasibility performance with filtering $\phi(Y)$ and $\phi(t, Y)$. The trend of reward changing is similar to Figure 2, where with larger thresholds, the performance will first increase and then go down. Safety filtering aligns with the stabilization to increase the reward, but the noise from the model mismatch between the neural operator and real dynamics will make the performance collapse if the safety filtering is too strong. For the boundary feasibility, we can see that the average feasible steps keep going up as the threshold increases, showing that the finite-time convergence is more enhanced for the feasible trajectories. However, when the threshold becomes too large, e.g. $\eta = 10$, the feasible rate also decreases significantly because the system is no longer stable, as the reward indicates.

**More comparison with different operators.** In this section, we show the comparison of two neural operators, FNO Li et al. (2020a) and MNO Li et al. (2022) for the safety filter performance with $\phi(Y)$ in learning the boundary mapping from control input $U(t)$ to output $Y(t)$ for 1D hyperbolic equation. Note that MNO models have larger model complexity than FNO models. Different from Table 5, in Table 7, we can see that with weaker BCBF $\phi(Y)$, MNO performs no worse than FNO in feasible rate and reward, showing that larger model complexity will compensate the performance of BCBF in the safety filter framework.

**More visualization of hyperbolic and Navier-Stokes equations.** Here we visualize the trajectories under 1D hyperbolic equation using SAC controller without and with safety filtering of $\phi(t, Y)$, as shown in Figure 4. Similar to 3, for each trajectory, the state value $u(x, t)$ after filtering is lower than that before filtering, i.e., the blue area is lower than the red area. For the output boundary, the filtered one $Y(t)_{\text{safe}}$ in blue solid lines goes towards the constraint $Y(t) < 0$ compared to the nominal boundary output $Y(t)_{\text{nominal}}$ in red solid lines, because of the output boundary. The difference is not very large in the last two figures because the threshold is relatively small to keep the stability of the output. As the visualization shows in Figure 5, it can be seen that the mid-upper high-speed tracking performance is improved compared to the baseline without filtering due to the constraint satisfaction. However, since the output boundary is just one point in the high-speed part, the general performance after filtering is not improved significantly, which is consistent with the findings in Table 3.

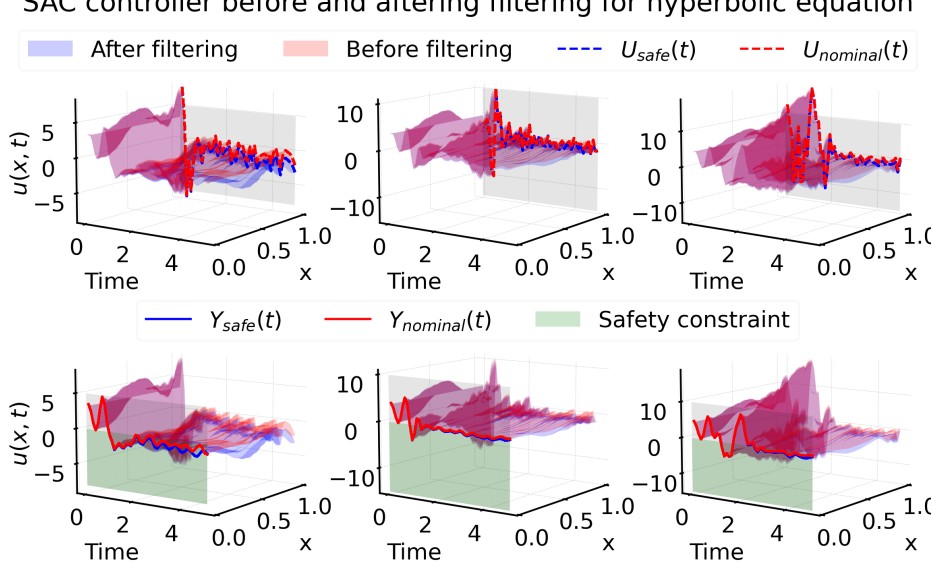

Figure 4: Visualization of state $u(x,t)$ of hyperbolic equation under SAC controller with (in blue) and without (in red) filtering. Boundary control inputs $U(t)$ are in dashed lines and boundary output $Y(t)$ are in solid lines. The boundary constraint $Y(t) < 0$ is in green.

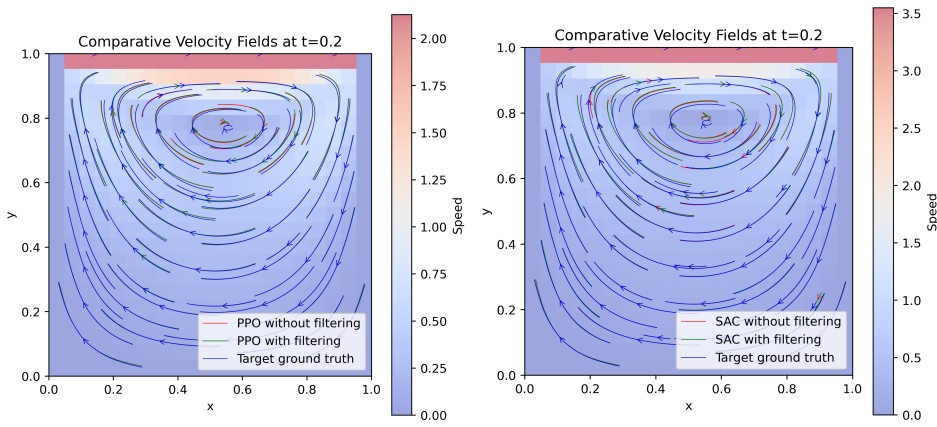

Figure 5: Visualization of tracking performance with PPO and SAC models before and after filtering with $\phi(t, Y)$ at the end time step of the trajectory for Navier-Stokes equation.

## C  LIMITATION AND DISCUSSION

Since the proposed method is based on neural operator modeling instead of real PDE dynamics, it does not directly solve the problem of model mismatch which may hurt the safety filtering performance in the implementation. We mark this important point as future work. Also, for PDE dynamics with higher-dimensional states, it is future work to investigate how BCBF can deal with spatially-dependent boundaries. Another limitation lies in that we do not adopt online safety filtering under the real PDE dynamics due to the delay of QP, and it is promising to improve it by filtering-induce policy which is found offline. It is also interesting to omit the iterative filtering by prediction using the one-time filtering for the whole trajectory based on Equation (10), which owns challenges of the nonlinear dependence of neural operator derivative at initial time.

