# OpenReview forum: "Guaranteed Neural PDE Boundary Control with Neural Barrier Function"
_ICLR.cc/2025/Conference — ICLR 2025 Conference Withdrawn Submission_

### Official Review · Reviewer_Pg7K · 2024-11-01

**Soundness:** 2
**Presentation:** 4
**Contribution:** 2
**Rating:** 3
**Confidence:** 5

**Summary:**

This paper presents a CBF-based approach to design boundary input so that the boundary output is safe.

**Strengths:**

+The key module in this paper is to learn the transition function from boundary input to boundary out, i.e., $Y(t)=G(U)(t)$, so that the model-based CBF approach is applicable.

**Weaknesses:**

-With the model learned, it falls within a standard CBF problem. It does have the time component in the CBF. However, time-dependent CBF has already been well studied, such as references listed in the paper, e.g.,  Dawson et al. (2023),  Garg & Panagou (2021b), and many others studying CBF + temporal logics. Then other than the specific model learning parts, all other core parts such as the loss functions and QP of CBF, can be readily found in the literature.

-The non-markovian property of boundary control is discussed. This paper proposes to explicitly use time $t$ as an argument to learn the input-output map $Y(t)=G(U)(t)$. However, this does not really address the non-Markovian issue, as it is still a point-to-point map at time $t$.

**Questions:**

In equation (7), "$]_+$" is missed.

The QP in (13) seems to be a single-variable convex problem that can be solved analytically. It is unclear why it cannot be solved in real-time.

Learning the model $Y(t)=G(U(t))$ is generally hard, and sample-complex for PDEs. It will be interesting to see how the learned models behaviors for high-dimensional PDE beyond 1-D and 2-D.

In Section 4.1, how is the data $(U(t), Y(t))$ collected? by solving the PDE? That might be computationally expensive and inapplicable for high-dimensional PDEs

The experimental results are confusing to interpret/explain, listed as follows.

In Table 1, the PPO baseline without a safety filter does not achieve the highest reward, which does not read right. Also, the rewards in this table are super noisy with large std. Considering that all three approaches (either the PPO ones or SAC ones) do not show a clear margin between them. That is probably why PPO/ SAC without safety filters do not achieve the highest reward. So I am wondering why the reward is so noisy.

Also in Table 1, for PPO approaches, why did the time-independent filter $\phi(Y)$ failed to improve the feasible rate? For SAC, the time-independent filter $\phi(Y)$ even lowers the feasible rate significantly, casting doubts on the effectiveness of this approach, or at least, the implementation.

Problems similar to those in Table 1 exist in Tables 2 and 3.

In Table 3, the tradeoff between reward and feasibility is not significant. It reads like without any loss of high reward, the feasible rate is improved by 13 percent.

It is not clear why "Partial derivative equations (PDEs)" is used while "partial differential equation" sounds more common.

**Details Of Ethics Concerns:**

None.

---

### Official Review · Reviewer_jcb1 · 2024-11-02

**Soundness:** 3
**Presentation:** 3
**Contribution:** 2
**Rating:** 3
**Confidence:** 3

**Summary:**

This paper presents a method for safe control of unknown partial differential equations, first learning the PDE dynamics using a neural operator, then using that operator along with a learned barrier function to filter a control input to safely control the system. Safety is defined as reaching a goal set in finite time and remaining there.

**Strengths:**

The paper is clearly written and easy to follow.
The authors make an interesting connection between PDE control and neural certificates, using the learned operator to reduce the complexity of the control problem. The overall approach is straightforward and easy to follow.

**Weaknesses:**

My main concerns with this paper are with its evaluation.

First, as discussed in the points below, the authors make several claims that are not supported by the provided data, and they do not provide other pieces of information (error bars on the safety rate) that would help in assessing their claims.

Second, the paper lacks a credible baseline. The authors compare an RL controller that was not trained with safety constraints against their filtered controller. Are there other safe PDE control strategies you could compare against? Perhaps re-training the RL controller with a safety term, or using a constrained RL algorithm rather than PPO or SAC? Or using a model-based controller with the learned neural operator model.

Generally, the point of the evaluation section is to help your reader understand why they should use your approach over some pre-existing or less complicated method. If you were to provide a stronger evaluation, I think this would be a very interesting paper.

Another major concern is that the authors do not provide a good explanation for why their CBF-based method does not achieve higher safety rates. You claim "guaranteed" control in the title, but the safety rate is only 71% on some problems. Why is that?

Specific concerns:
- No error bars are reported for feasible rate or average feasible steps. This could be done by running 10 sets of 100 episodes and reporting mean and std. deviation across those 10.
- In Tables 1, 2, 4, and 5, none of the rewards should be in bold, since they are all within 1 standard deviation of each other (you don't have statistical evidence to claim that any of these methods achieves a higher reward). This is also true for the SAC results in Table 3.
- Page 8, Line 412: I don't see this plot showing reward "increasing slightly" given the error bars.
- Page 9: "FNO presents higher rewards under both PPO and SAC base models, showing that FNO is more suitable for learning low-resolution trajectories with 50 sampled points." Are you really able to make this claim? These two rewards are indistinguishable given the reported error bars.

Minor issues to change
- Consider using `\citep` to put citations in parenthesis.

**Questions:**

1. The intuition for including the filtering threshold $\eta$ is not clear to me, as it would seem to allow unsafe behavior when $\dot{U}_{nominal}$ is far from being safe. Could you please explain why you include this threshold?
2. You say that you collect training data using the pre-trained RL controllers. Is there any risk that replacing the nominal controller with the filtered controller will cause a distribution shift that creates an error in your learned transfer function & BCBF? How would you mitigate this error? Could you instead use some other control input (e.g. persistently exciting) to gather training data?
3. In Tables 1 and 2, why did you use different boundary feasibility constraints for PPO and SAC?
4. Could you comment on why you think the feasibility rate is not 100%? Is the BCBF not fully trained? Is there model mismatch?
5. "because time-independent BCBF tends to have divergent performance with more non-feasible trajectories and more feasible steps for feasible trajectories" -> why do you think this is?
6. "showing that the strong safety filtering may hurt the stability of the PPO controller due to the model mismatch between direct boundary mapping with the neural operator and underlying PDE dynamics" -> Is this because the neural operator is trained only using the nominal controller? Could you learn these dynamics using a random control input and get better results with less mismatch? How do you know that filtering hurts the stability of the controller; could it instead be that high reward is not compatible with feasibility on this problem?

---

### Official Review · Reviewer_3ohc · 2024-11-02

**Soundness:** 3
**Presentation:** 3
**Contribution:** 3
**Rating:** 6
**Confidence:** 1

**Summary:**

N/A.

**Strengths:**

N/A.

**Weaknesses:**

N/A.

**Questions:**

N/A.

---

### Note · Authors · 2024-11-26

I have read and agree with the venue's withdrawal policy on behalf of myself and my co-authors.